# Portosystemic shunting prevents hepatocellular carcinoma in non-alcoholic fatty liver disease mouse models

Andrea Peloso[1,2]ᵒ*, Stéphanie Lacotte[2]ᵒ, Quentin Gex[2], Florence Slits[2], Beat Moeckli[2], Graziano Oldani[1,2], Matthieu Tihy[3], Aurélie Hautefort[4], Brenda Kwak[4], Laura Rubbia-Brandt[3], Christian Toso[1,2]*

1 Division of Abdominal Surgery, Department of Surgery, Geneva University Hospitals and Faculty of Medicine, Geneva, Switzerland, 2 Transplantation and Hepatology Laboratory, University of Geneva, Geneva, Switzerland, 3 Division of Clinical Pathology, Geneva University Hospitals and Faculty of Medicine, Geneva, Switzerland, 4 Department of Pathology and Immunology, University of Geneva, Geneva, Switzerland

ᵒ These authors contributed equally to this work.
* andrea.peloso@hcuge.ch (AP); christian.toso@hcuge.ch (CT)

**Data Availability Statement:** The raw data required to reproduce the above findings are available to download from https://yareta.unige.ch/

## Abstract

### Background and aims

Non-alcoholic fatty liver disease (NAFLD) is one of the leading cause of hepatocellular carcinoma (HCC). This association is supported by the translocation of bacteria products into the portal system, which acts on the liver through the gut-liver axis. We hypothesize that portosystemic shunting can disrupt this relationship, and prevent NAFLD-associated HCC.

### Methods

HCC carcinogenesis was tested in C57BL/6 mice fed a high-fat high-sucrose diet (HFD) and injected with diethylnitrosamine (DEN) at two weeks of age, and in double transgenic LAP-tTA and TRE-MYC (LAP-Myc) mice fed a methionine-choline-deficient diet. Portosystemic shunts were established by transposing the spleen to the sub-cutaneous tissue at eight weeks of age.

### Results

Spleen transposition led to a consistent deviation of part of the portal flow and a significant decrease in portal pressure. It was associated with a decrease in the number of HCC in both models. This effect was supported by the presence of less severe liver steatosis after 40 weeks, and lower expression levels of liver fatty acid synthase. Also, shunted mice exhibited lower liver oxygen levels, a key factor in preventing HCC as confirmed by the development of less HCCs in mice with hepatic artery ligation.

### Conclusions

The present data show that portosystemic shunting prevents NAFLD-associated HCC, utilizing two independent mouse models. This effect is supported by the development of less

home/detail/1efd54be-6acd-4117-9fbe-f1ca2ac88674 https://doi.org/10.26037/yareta:qopvcbpslnfjvllbbngso3nlqy).

**Funding:** The Swiss National Science Foundation (grant number 182471), the "Fondation Francis & Marie-France Minkoff", the "Fondation de la Recherche Médicales Carlos et Elsie De Reuter", the "Fondation Gilles Mentha", the Swiss Society of Gastroenterology (SGG-SSG), the Swiss Society of Visceral Surgery (SGVC-SSCV), and the Leenaards Foundation (grant number 5489) funded this research. The funders had no role in study design, data collection and analysis, decision to publish, or preparation of the manuscript.

**Competing interests:** The authors have declared that no competing interests exist.

steatosis, and a restored liver oxygen level. Portal pressure modulation and shunting deserve further exploration as potential prevention/treatment options for NAFLD and HCC.

## Introduction

Hepatocellular carcinoma (HCC) is the third leading cause of cancer-related mortality worldwide [1].

It is also the cancer with the fastest increasing incidence in many Western countries due to its association with non-alcoholic fatty liver disease (NAFLD) [2].

The gut-liver axis has been shown to play a significant role in the development of HCC in the setting of NAFLD. Among others, blocking the lipopolysaccharide (LPS)- toll-like receptor 4 (TLR4) pathway, prevents the action of bowel-released bacterial products on the liver, and prevents the initiation and growth of HCC [3, 4]. The porta acts as a key anatomical link in this relationship.

NAFLD is associated with portal hypertension even in patients without fibrosis, driven by hepatocyte ballooning, endothelial cell dysfunction, and liver sympathetic nerve alterations [5]. Portal hypertension does alter the gut permeability and increases the release of bacterial products into the portal system, which ultimately acts on the liver, worsening the state of NAFLD [6, 7]. This multi-step relationship creates a vicious circle, also leading to HCC as a known complication of NAFLD.

A portosystemic shunt is an abnormal connection between the portal and systemic vascular systems. Blood from the abdominal organs, which should be drained by the portal vein into the liver, is instead diverted to the systemic circulation with a concomitant decrease in portal pressure [8]. Transjugular intrahepatic portosystemic shunt (TIPS) has been successfully used to treat portal hypertension in patients. However, the impact of shunting on HCC remains unclear [9], but one can hypothesize that it can break the NAFLD-portal hypertension-gut vicious circle and prevent HCC. We therefore study the impact of portosystemic shunting, utilizing and charactering a reproducible model of spleen transposition in mice with NAFLD-associated HCC. We also explore potential underlying mechanisms leading to the alteration of the risk of HCC after shunting, looking at the state of the underlying liver disease and its level of oxygenation. Such observations are important as they can bring more light on the steps leading to HCC and also on potential clinically-relevant interventions.

## Materials and methods

The raw data required to reproduce the above findings are available to download from https://yareta.unige.ch/home/detail/1efd54be-6acd-4117-9fbe-f1ca2ac88674 https://doi.org/10.26037/yareta:qopvcbpslnfjvllbbngso3nlqy).

### Mouse models and diets

The control diet (CD) was the standard chow (SAFE® 150, Safe Diets). NAFLD (steatosis) was induced by feeding four-week-old male C57BL/6J mice (Janvier Labs) a 45% kcal fat, 40% kcal carbohydrate diet containing 0.05% cholesterol (ENVIGO TD.08811). For chemical induction of HCC, C57BL/6J mice were intraperitoneally injected with 25 mg/kg N- nitroso-diethylamine (DEN, Sigma-Aldrich) at 14 days of age. The mice were fed a high-fat high-sucrose diet (ENVIGO TD.08811) starting from 5 weeks of age. In the genetic model of HCC, the double transgenic LAP-tTA and TRE-MYC mice (LAP-Myc) were fed a doxycycline-

enriched diet until four weeks of age (SAFE® 150 0.625g/Kg Doxycycline Hyclate). Thereafter, they were fed methionine-choline-deficient (MCD) diet (Safe Diets, U8958 Version 348). This model was a kind gift of Prof. T. Greten and has been described previously [10].

## Experimental design and animal groups

We used two different models to test the impact of portosystemic shunting on NAFLD-associated HCC carcinogenesis. First, HFD-fed C57BL/6 mice were injected with DEN at two weeks of age and underwent spleen transposition at eight weeks of age (**DEN/shunt**). The control HFD-fed C57BL/6 mice were injected with DEN, but underwent a sham surgery (**DEN**). Second, MCD-fed LAP-Myc mice underwent spleen transposition at eight weeks of age (**LAP-Myc/shunt**). MCD-fed LAP-Myc mice with sham surgery (**LAP-Myc**) were used as a control.

To test the impact of portosystemic shunting on NAFLD, we used HFD-fed male C57BL/6 mice which underwent spleen transposition at eight weeks of age (**HFD/shunt**). The control group consisted of HFD-fed C57BL/6 mice which underwent a sham surgery (**HFD**).

To study the role of the hepatic artery, HFD-fed C57BL/6 mice were injected with DEN at two weeks, underwent spleen transposition at eight weeks and hepatic artery ligation at ten weeks (**DEN/shunt/Art-**). Control C57BL/6 mice were fed HFD and injected with DEN, with spleen transposition, but without hepatic artery ligation (**DEN/shunt/Art+**).

## Creation of portosystemic shunts by subcutaneous spleen transposition

We used an adapted version of a rat protocol introduced by Bengmark *et al*. [11]. Mice were anesthetized under 5% isoflurane, subcutaneously injected with buprenorphine (0.1 mg/kg) and maintained under 2% isoflurane during the time of the protocol.

A 7 mm left subcostal laparotomy was performed and a subcutaneous pouch was created. The spleen was pulled out and its surface scratched with the tip of a needle (25G x5/8"; BD Microlance, Becton, Lout, Ireland) until gentle venous bleeding was induced (seven to ten longitudinal scrapes were required to obtain a satisfactory result). The spleen was then placed in a subcutaneous pocket. One stitch (Prolene 8/0) was placed around the muscular neck of the pouch to prevent the spleen from moving back into the abdomen, and the skin was closed over the spleen. In sham-operated animals, the spleen was pulled out and then reinserted into the abdomen before the incision was closed. Mice were injected again with buprenorphine (0.1 mg/kg) 8 hours after the surgery. Animals were monitored according to a detailed score sheet with specific humane endpoints (weight loss $\geq$ 15% of initial weight, inactive, hunched posture). At time of sacrifice, the same procedure was used.

## Portal pressure measurements

After 40 weeks of HFD, we measured portal pressure in sham-operated and shunted mice (n = 4 group CD; n = 10 group HFD; and n = 10 group HFD/shunt). After a midline laparotomy, the ileocolic vein was cannulated using 27G BD Valu-Set (BD, Belliver Industrial Estate, Plymouth, UK) and connected to a calibrated digital pressure analyser (Micro Pressure Transducer with Amplifier; Radnoti Monrovia, CA, US). Portal pressure was continuously monitored, and the average over 30 seconds was recorded. Upon sacrifice of the animal, the spleen and the liver were removed, weighed, and liver/spleen weight ratio was calculated.

## Quantification of the portosystemic shunt permeability

The venous shunt permeability was assessed by injecting $5 \times 10^4$ microspheres (FluoSphere<sup>TM</sup> polystyrene, 15 µm, scarlet [wl 645/680]; Invitrogen by Thermo Fisher Scientific, Waltham,

US) in 50 µl of the microsphere carrier solvent, through the ileocolic vein. 5 minutes after the injection, the mice (n = 5 group HFD; n = 8 group HFD/shunt) were sacrificed. We then measured the fluorescence intensity emitted by microspheres with the intravital system IVIS-200 detector (Xenogen, Alameda, CA, US) in the explanted livers and spleens. The ratio of the hepatic to splenic signal was quantified using the following formula: [liver fluorescence/(spleen fluorescence + liver fluorescence)] [12].

## Liver histology evaluation

Liver sections were blindly reviewed by an expert pathologist in order to assess the SAF score [13] and to classified the nodules based on morphologic criteria [14]. In order to quantify the steatosis, hematoxylin and eosin (H&E) stained liver sections were acquired on an Axioscan Microscope Slide Scanner (Zeiss, Oberkochen, Germany) and analysed with QuPath (https://qupath.github.io/). The whole analysis process is described with the raw data.

## *In vivo* micro-CT imaging, liver and spleen volumetry and tumour quantification

*In vivo* imaging was performed using a micro-CT system (Quantum GX, PerkinElmer, Hopkinton, MA, USA) after intravenous injection of an angiography contrast agent (ExiTron nano 12000, 100µl/animal; Miltenyi Biotec, Bergisch-Gladbach, Germany) and with a respiratory gating strategy [15]. The analysis of the images was performed using OsiriX (v.12.0 www.osirix-viewer.com). All *in vivo* imaging procedures were performed under isoflurane anaesthesia.

All animals with spleen transposition underwent a contrast-enhanced CT at the end of the experiment to confirm the presence of a venous shunt. The scan was performed 2 minutes after contrast injection. Coronal, sagittal, and axial reconstructions were prepared and analysed to establish 2D maximum intensity projection. In the case of a missing or incomplete portosystemic shunt, the animal was excluded from the study. Hepatic and splenic volumes were calculated based on a second CT-scan performed 24 hours after contrast injection. The volumetric assessment was performed on sets of axial images using slice thickness of 1–2 mm and contouring the liver and spleen as regions of interest. Hepatic and splenic volumes were then calculated automatically. Each calculation was carried out in triplicate and then averaged. Tumour burden was determined by counting tumour nodules and manually tracing their contours in each slice, and covering the whole liver.

## Ex *vivo* MRI and tumour quantification

DEN-induced tumorigenesis after 40 weeks was assessed by micro-MRI performed on explanted livers, as previously described [16]. Livers were analysed with a micro-MRI scanner (Nanoscan 3T, RS2D, Mundolsheim, France) with a birdcage coil of 3.5 cm in diameter. After automatic adjustment of the B0 homogeneity, T1w and T2w images were acquired. Tumour nodules were measured in OsiriX (v.12.0 www.osirix-viewer.com) as described above.

## Hepatic gene expression

After total RNA extraction of liver tissue (ReliaPrepTM RNA Tissue, Promega), cDNA was synthesised by extending a mix of random primers with the High Capacity cDNA Reverse Transcription Kit in the presence of RNAase Inhibitor (Applied Biosystems). The relative quantity of each transcript was normalized to the expression of EEF1, HPRT and GAPDH. SYBRGreen reagent was used for Real-time PCR on the ABI Prism 7000 sequence detection

system (Applied Biosystems) according to the manufacturer's instructions. Primer sequences are provided in S1 Table.

## Hyperspectral evaluation of hepatic and intestinal oxygenation

After a midline laparotomy, the liver and the small bowel were completely exposed, and hyperspectral images were acquired. The hyperspectral camera acquires 2D images with spectroscopic information at a determined wavelength as the third dimension. In our study, a commercially available hyperspectral camera (TIVITA®, Diaspective Vision GmbH, Germany) was used to quantify tissue oxygenation through the emission and the absorbance in the visible and near-infrared (NIR) electromagnetic spectrum (400–1000 nm). The distance between the camera and the organ was 40 cm.

## Vascular function

Vascular reactivity was assessed using a wire myograph (Multi wire Myograph System, Model M620, DMT) as previously described [17]. Briefly, mice (CD, HFD and HFD/shunt groups) were sacrificed, their thoracic aorta segments quickly removed (length: 1.5–2 mm) and mounted on two 40-μm diameter tungsten wires. The vessels were preserved in an organ bath containing bubbled (95% $O_2$ and 5% $CO_2$) physiological Krebs solution (115 mM NaCl, 2.5 mM $CaCl_2$, 4.6 mM KCl, 1.2 $KH_2PO_4$, 1.2 mM $MgSO_4$, 25 mM $NaHCO_3$ and 11.1 mM glucose) at 37˚C.

The blood vessels were normalized and the inner circumference corresponding to 90% the outer circumference at transmural pressure of 100 mmHg was measured. After 45-minute stabilization, the vessels were constricted in 100 mM $K^+$ solution. We measured dose response of vasoconstriction to KCl and U46619 (Thromboxane A2 agonist) and of vasodilation to acetylcholine and sodium nitroprusside (endothelium-dependent and -independent response, respectively) after U46619 pre-contraction. Data were analysed with LabChart software (ADInstruments, Sydney, Australia).

## Bile acids analysis

Bile acid analysis was performed by the Metabolomics Unit at UNIL, Switzerland. Briefly, portal bile acids were extracted from 20 μL of plasma with 80 μL of ice-cold methanol (n = 8 group CD; n = 13 group HFD; n = 12 group HFD/shunt) and analysed by reversed phase liquid chromatography coupled to high resolution mass spectrometry (QExactive Focus—Thermo Fisher Scientific). Raw data was processed using MS-Dial software [18]. Peak areas were normalized to the sample amount (mg). Signal intensity drift was corrected in R with the LOWESS/Spline normalization program followed by noise filtering and visual inspection of linear response.

## Hepatic artery ligation

To verify the effect of the liver artery perfusion on HCC carcinogenesis, we performed common hepatic artery ligation or sham surgery in DEN-treated shunted mice (n = 13 DEN/shunt/Art-; n = 13 group DEN/shunt/Art+). Under isoflurane anaesthesia, a midline laparotomy was performed to expose the hepatic hilum. The common hepatic artery was isolated and ligated by tying two monofilament nylon ligatures (Ethilon 8/0; Ethicon US, LLC, USA). A 1-mm segment of the hepatic artery between the two ligatures was then removed. The control mice underwent similar surgical procedure, with the hepatic pedicle exposed and mobilized but without hepatic artery ligation. The abdominal wall was closed with running sutures (Prolene 6/0).

## Ethics

Animal experiments were performed in compliance with the international regulations and following research protocols approved by the University of Geneva Institutional Animal Care Committee and Geneva veterinary authorities (authorization numbers GE/167/17 and GE/2A). All animals received humane care compliant with the criteria outlined in the "Guide for the Care and Use of Laboratory Animals". Mice were given free access to food and water, and were housed in a temperature-controlled facility with a 12-hour light/dark cycle. Fresh chow was provided weekly. Food intake and body weight were assessed weekly. After surgery, mice were allowed to recover on a heating pad and were given subcutaneous buprenorphine (0.05 mg/kg).

## Statistical analysis

Statistical analyses were performed using GraphPad Prism software 8.3.1 (Software Inc., La Jolla, CA). Data are expressed as median ± interquartile range. Differences were considered significant at <0.05. Comparisons between three groups were also analysed using a Kruskal-Wallis test and were all significant. Comparisons between groups were performed using the Mann-Whitney U and were shown in the figures.

## Results

### Portosystemic shunting reduces portal pressure

All animals survived the spleen transposition surgery. Seven weeks after spleen transposition, the spleno-axillary venous shunts were visible on CT scans (Fig 1A) in 83.3% of mice. The persistence of the shunt was also confirmed at 40 weeks with a second *in vivo* CT scan and macroscopic observation (Fig 1A).

We evaluated shunt development and function by measuring the liver/spleen luminescence ratio after injection of fluorescent microbeads into the ileocolic vein. A significantly higher ratio was detected in non-shunted compared to shunted HFD mice (p = 0.02; Fig 1B). This confirms that the shunt partially diverted the portal blood away from the liver and increased an hepatofugal splenic vein flow. Given our hypothesis that shunting impacts portal haemodynamics, we studied the effect of shunting on portal pressure. After 40 weeks of HFD, the mice had almost double the pressure of the CD-fed mice (p = 0.001). Shunting decreased portal pressure (p<0.0001) to levels close to those of control mice (Fig 1C).

Portosystemic shunting can lead to some degree of encephalopathy [19], and we were concerned that this could affect food intake and, consequently, the obesogenic effects of HFD. We therefore demonstrated that the average daily food intake (Fig 1D) and movement pattern (data available in the Yareta repository (MasterFile)) were similar in the two groups. In addition, shunted and non-shunted mice exhibited similar weight gain (Fig 1E).

A previous study has shown that NAFLD may be closely associated with atherosclerotic vascular disease in addition to other well-known cardiovascular risk factors [20]. Venous shunting did not alter the vascular reactivity of the aorta; similar dose responses were observed upon KCl- and U46619-induced vasoconstriction as well as upon endothelium-dependent (acetylcholine) and -independent (sodium nitroprusside) dilation (S1 Fig). Venous shunting can lead to chronic inflammation related to the lack of the hepatic first pass. We showed that shunting did not affect the heart rate and the blood pressure profiles (S2 Fig).

Taken together, these results show that shunting decreases portal hypertension in a long-term HFD-fed mouse model, without altering systemic physiological and metabolic aspects.

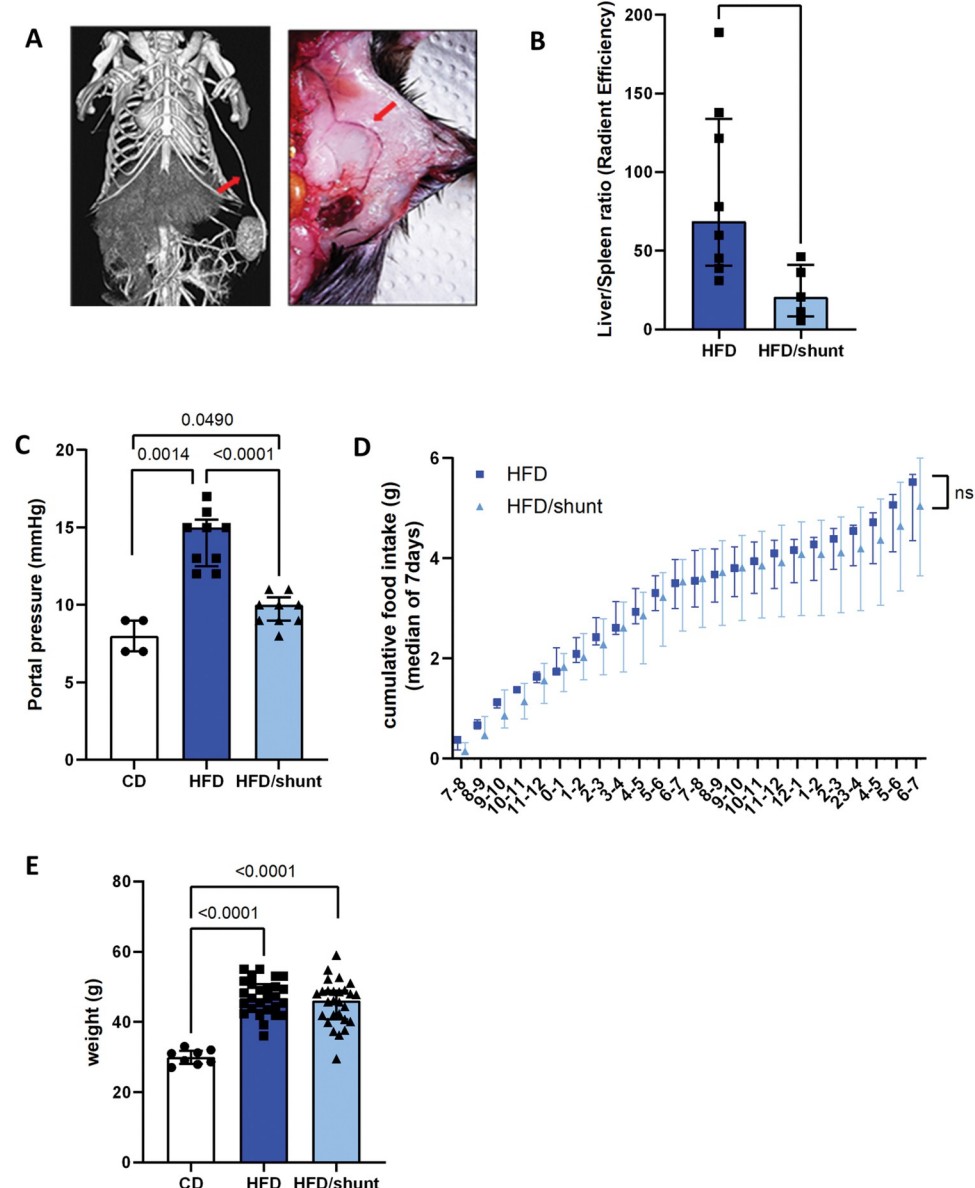

**Fig 1. Effect of portosystemic shunting on portal pressure, food intake, body weight and body composition in high-fat diet-fed C57BL/6 mice.** Spleen transposition resulted in portosystemic venous shunting (A). Its effect was assessed by the liver/spleen fluorescence ratio after injection of fluorescent microbeads in the ileo-colic vein (HFD, n = 8, HFD/shunt, n = 5) (B). The portal blood pressure was assessed in control diet mice (CD, n = 4), HFD-fed mice (HFD, n = 9) and HFD-fed mice with shunt (HFD/shunt, n = 9) (C). The cumulative food intake was determined by evaluating the food intake per hour for 7 days in 40 weeks old HFD (n = 6) and HFD/shunt mice (n = 5) (D). Body weight was measured in CD, HFD and HFD/shunt groups at 40 weeks (CD, n = 8, HFD, n = 27, HFD/shunt, n = 27) (E). Data are expressed as median ± IQR. Comparisons between groups were performed using the Mann-Whitney U.

## Portosystemic shunting prevents HCC carcinogenesis

We explored whether shunting affects HCC carcinogenesis in the DEN-induced HCC mouse model. Shunting reduced the number of HCCs (p = 0.047; Fig 2A, S3 Fig and S1 File). In an independent set of experiments with CT scans at 28 and 36 weeks, we confirmed that shunting is associated with fewer HCCs especially at the later time point (reduced carcinogenesis) (Fig 2B

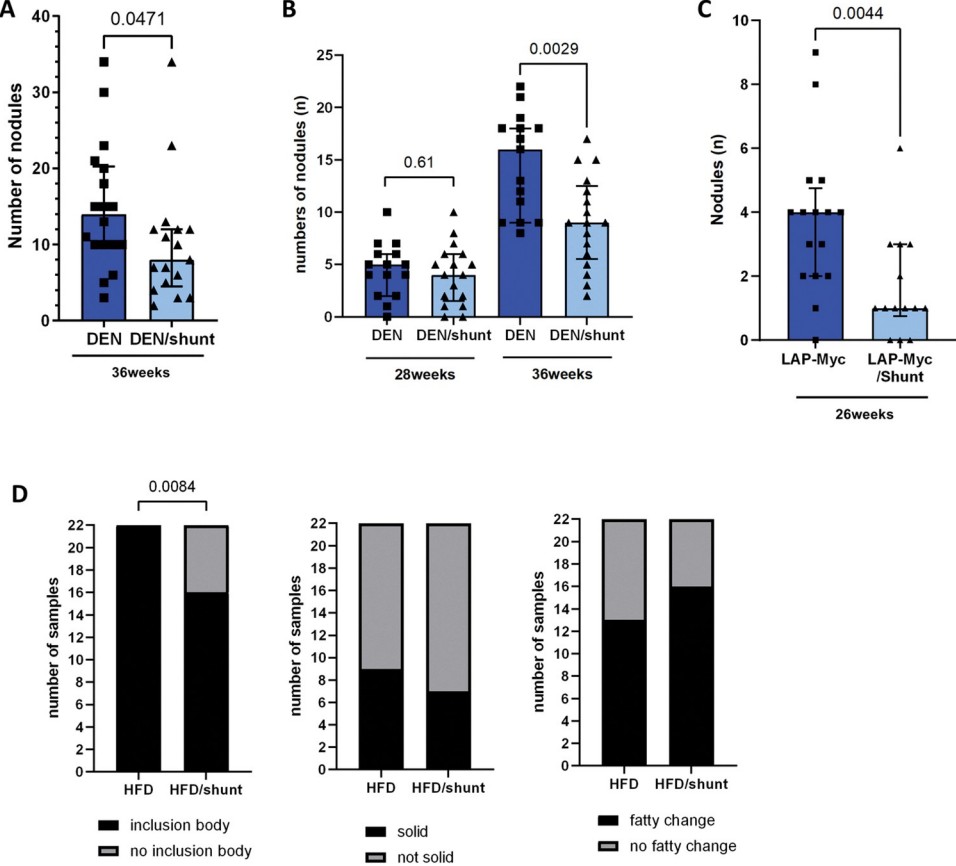

**Fig 2. Effect of portosystemic shunting on HCC carcinogenesis.** Via MRI images, tumor burden was assessed at week 40, looking at tumor count (A) in DEN-induced (DEN, n = 18) and DEN-induced mice with shunt (DEN/shunt, n = 17). In an independent experiment, tumor number was assessed by CT scan at weeks 28 and 36 in DEN (n = 15) and DEN/shunt (n = 17)(B). HCC count was also assessed in LAP-Myc HCC transgenic mice at weeks 20 and 26 (LAP-Myc versus LAP-Myc/shunt)(C). Assessment of three parameters characteristic of tumor morphology: Presence of inclusion body, solid growth pattern and fatty change on 22 lesions from HFD-fed mice and 22 lesions from shunted HFD-fed mice (D). Data are expressed as median ± IQR. Comparisons between groups were performed using the Mann-Whitney U or the Chi-square.

and S3 Fig). We also explored the effect of shunting on carcinogenesis in the LAP-Myc HCC transgenic mouse model. Again, we confirmed that shunting reduces carcinogenesis (p = 0.004; Fig 2C). DEN mice model are known to develop tumors and dysplasic nodules [14]. In order assess the pathological status of the nodules, we analysed lesions from DEN-HFD and DEN-HFD/shunt livers. Only one nodule was not characterized as HCC (1/44). In addition, three parameters characteristic of tumour morphology were assessed: the solid growth pattern, the presence of inclusion bodies and the fatty change. A small number of tumours did not include inclusion bodies similar to Mallory-Denk bodies in HFD/shunt mice (Fig 2D).Overall, shunting reduced carcinogenesis in three independent sets of experiments using two different HCC mouse models.

## Portosystemic shunting protects from hepatic steatosis

In an effort to explore the mechanisms underlying the decreased HCC carcinogenesis, we studied the impact of shunting on NAFLD. While steatosis can lead to hepatomegaly [21], we first used liver weight and volume as surrogate markers of NAFLD. As expected, HFD-fed

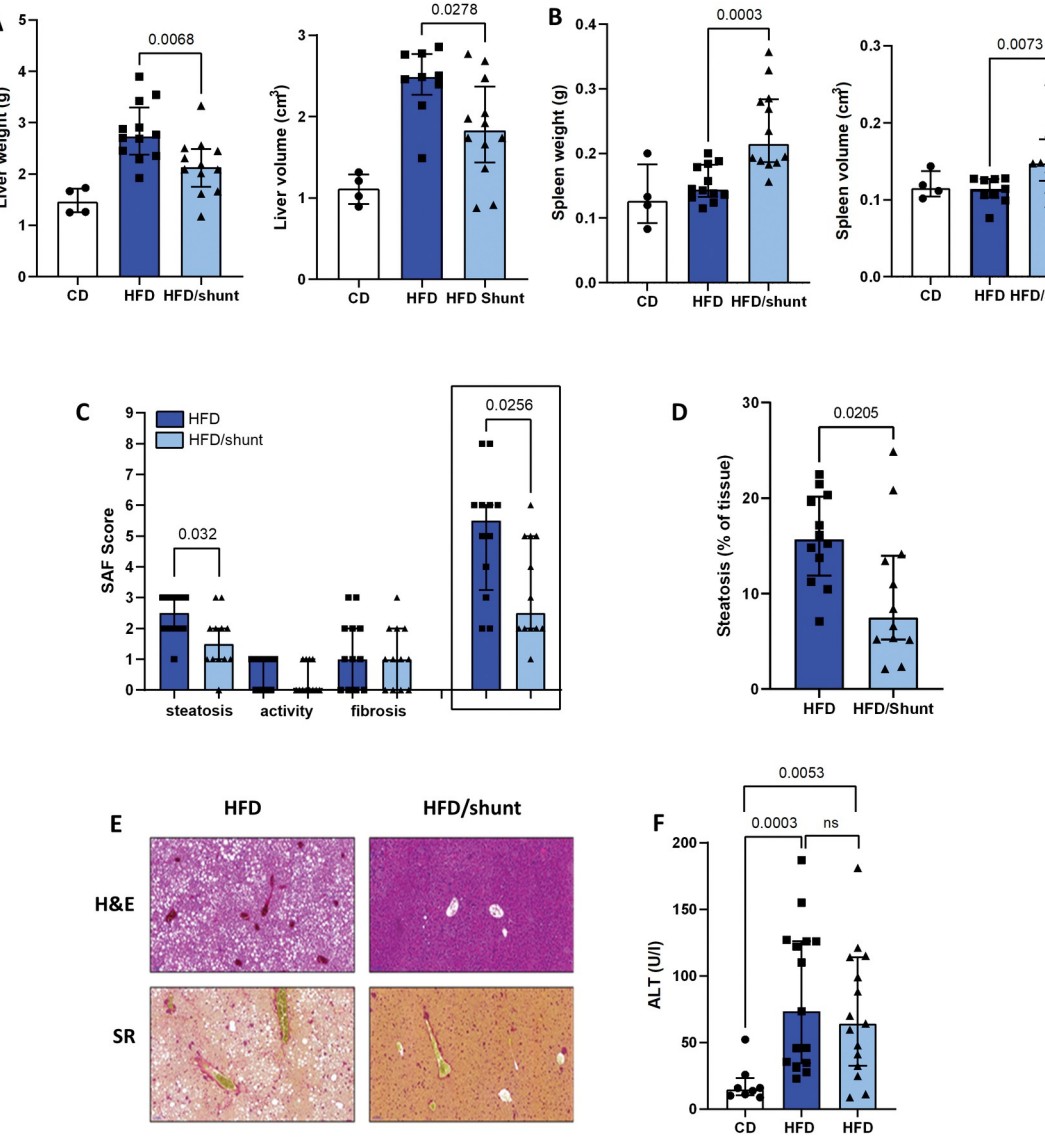

**Fig 3. Effect of portosystemic shunting on liver steatosis.** Compared to control diet mice (CD), a 40-week HFD induced hepatomegaly and liver weight increase, which were prevented by shunting (HFD/shunt) (CD, n = 4; HFD, n = 12; HFD/shunt, n = 12)(A). Spleen weight and volume (CD, n = 4; HFD, n = 12; HFD/shunt, n = 12) (B). NAFLD-Activity Score (HFD, n = 12; HFD/shunt, n = 12) (C) comprising steatosis, activity, and fibrosis was determined from hematoxylin and eosin (H&E) and Sirius Red (SR) stained sections of mouse liver from HFD and HFD/shunt groups at 40 weeks (E). Automated quantitative analysis of steatosis was performed on H&E liver sections (D). Serum ALT level was profiled for CD (n = 7), HFD (n = 14) and HFD/shunt (n = 13) groups (F). Data are expressed as median ± IQR. Comparisons between groups were performed using the Mann-Whitney U.

mice showed a significant increase in liver weight compared to CD mice (p = 0.001). Shunting partially abolished this liver weight increase (p = 0.007; Fig 3A). This suggests that portosystemic shunting can protect against hepatomegaly. We further confirmed this by comparing liver volumes calculated based on CT scans (Fig 3A). To investigate the impact of portosystemic shunts on splenomegaly, we also determined spleen weight, spleen to body weight ratio and spleen volume. In shunted mice, in which the blood flow is diverted to the spleen, we observed that the spleens were heavier than in non-shunted mice (p<0.001; Fig 3B). Consistently, spleen

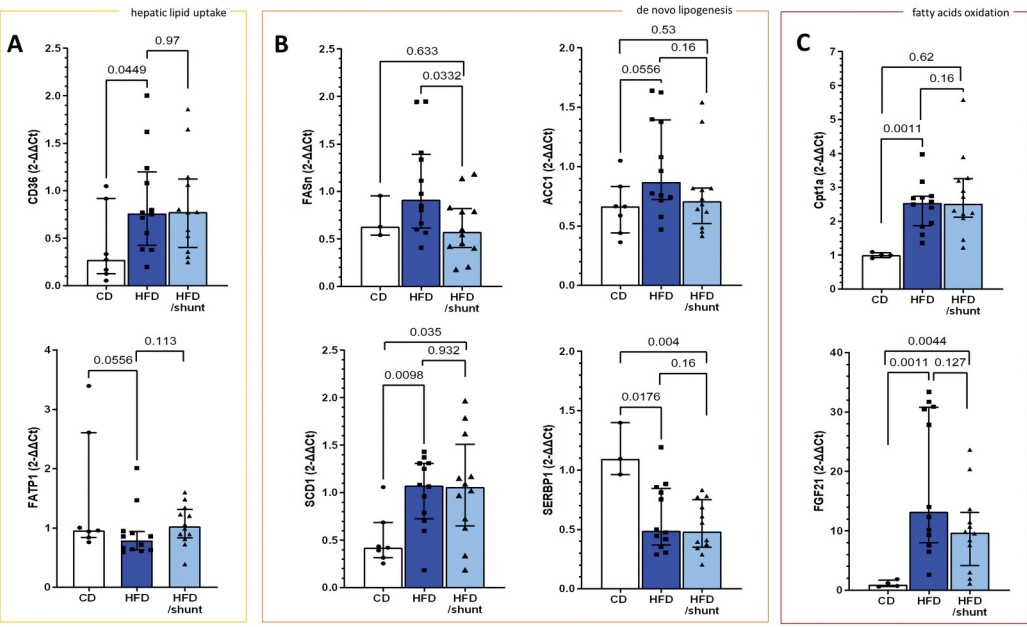

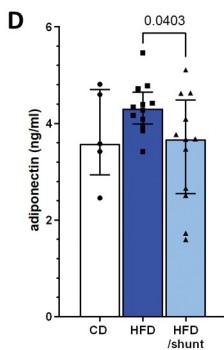

**Fig 4. Effect of portosystemic shunting on hepatic fatty acid metabolism.** Expression of genes involved in the hepatic lipid uptake pathway (A), de novo lipogenesis (B) and fatty acid oxidation (C) was profiled in control diet-fed (CD, n = 3), HFD-fed mice (HFD, n = 12) and HFD-fed mice with shunt (HFD/shunt, n = 12). Serum levels of adiponectin after the 40-week experimental period in CD (n = 5), HFD (n = 12) and HFD/shunt (n = 12) groups (D). Data are means as median of fold change ± IQR. Comparisons between groups were performed using the Mann-Whitney U.

volume and the ratio of spleen to the body weight were higher in shunted mice (p = 0.007 and p<0.001, respectively; Fig 3B and data available in the Yareta repository (MasterFile)).

Histological analysis revealed less steatosis in shunted compared to non-shunted HFD mice (p = 0.032; Fig 3C–3E). We did not observe significant differences in liver inflammation and fibrosis profiles (p = 0.41 and p = 0.81). Consistent with these data, serum ALT levels in non-shunted and shunted HFD mice were similar (p = 0.632, Fig 3F).

Hepatic steatosis results from the loss of liver lipid homeostasis, which is maintained through three major pathways: uptake of circulating lipids, *de novo* lipogenesis and fatty acid oxidation [22]. First, we confirmed that long-term (40 weeks) HFD altered gene expression in all three pathways—uptake of lipids (CD36 and FATP1, Fig 4A), *de novo* lipogenesis (FASn, ACC1, SCD1 and SREBP1, Fig 4B) and fatty acid oxidation (CPT1A and FGF21, Fig 4C). We studied the impact of shunting on each of these pathways. Shunts had no added effect on the

expression of the lipid uptake and fatty acid oxidation genes. The expression levels of the *de novo* lipogenesis pathway genes were partially restored after shunting, specifically, FASn was reduced to CD levels (p = 0.03; Fig 4B). In parallel, plasma adiponectin concentration was decreased in shunted HFD mice (p = 0.04; Fig 4D).

Collectively these findings indicate that shunting affects liver-spleen blood flow, and alleviates hepatic steatosis potentially acting through the *de novo* lipogenesis pathway, and leading to a liver more resistant to HCC carcinogenesis.

### Portosystemic shunting partially restores bile acid metabolism and is associated with a restored bowel lymphangiogenesis

Bile acid derivatives and compounds that influence bile acid-related signalling pathways are emerging as therapeutic targets for NAFLD [23], potentially regulating NAFLD progression and portal hypertension development [24]. Accordingly, we profiled the primary and secondary portal bile acid metabolites. As previously reported [25], we detected an important alteration in bile acid metabolism in HFD-fed compared to CD-fed animals (Fig 5A). Further analyses revealed a significant upregulation of the primary bile acids tauro-conjugated alpha- and beta-muricholic acids (T-α-MCA and T-β-MCA) in shunted compared to non-shunted HFD mice (p = 0.025 and p = 0.041, respectively) (Fig 5B and 5C). Shunting also restored physiological levels of taurohyodeoxycholic acid (THDCA) (p = 0.04; Fig 5D).

In the proximal bowel, the intestinal villi were longer in HFD-fed compared to CD-fed mice (p = 0.002; Fig 5E). There was no difference between shunted and non-shunted mice. Paneth cell/villus ratios were comparable in all groups (p = 0.329 and p = 0.082, respectively; Fig 5F). Goblet cells were less numerous in HFD mice (p = 0.004), with no additional impact of shunting (Fig 5G). Intestinal lymphatics are emerging as important player in the regulation of intestinal lipid mobilization and have an impaired integrity and permeability in metabolic syndrome [26]. We detected an increased number of lymphatic vessels in HFD-fed compared to CD-fed mice (p = 0.004), which was abolished by shunting (p = 0.017, Fig 5H).

Taken together, these data confirm the impact of HFD on the proximal small bowel and show that shunting is associated with a restored lymphangiogenesis.

### Portosystemic shunting affects liver oxygenation

To further identify factors underlying the decreased carcinogenesis [27], we investigated the hepatic oxygen delivery, as HCCs are highly oxygen-dependent with a vascular supply mainly coming from the hepatic artery [28]. We determined liver oxygenation levels using hyperspectral imaging (Fig 6). Compared to CD, HFD increased liver oxygenation (p = 0.0004), while shunting restored it to a close-to-normal level (p<0.001; Fig 6A).

To assess the impact of the shunt-associated decrease in liver oxygenation on carcinogenesis, we analysed the impact of hepatic artery ligation in the DEN-induced HCC mouse model. Hepatic artery ligation led to decreased carcinogenesis (group DEN/shunt/Art-, fewer HCCs) compared to mice with intact artery (group DEN/shunt/Art+, p = 0.026; Fig 6B). Hepatic artery ligation was also associated with smaller HCCs (p = 0.027; Fig 6C). In summary, these data demonstrate the impact of shunt on liver oxygen level, and the impact of the oxygen level on HCC carcinogenesis.

### Discussion

The present study further supports the impact of the gut-liver axis on the risk of HCC, using an innovative portosystemic shunt model in mice with NAFLD. Beyond the mechanistic link,

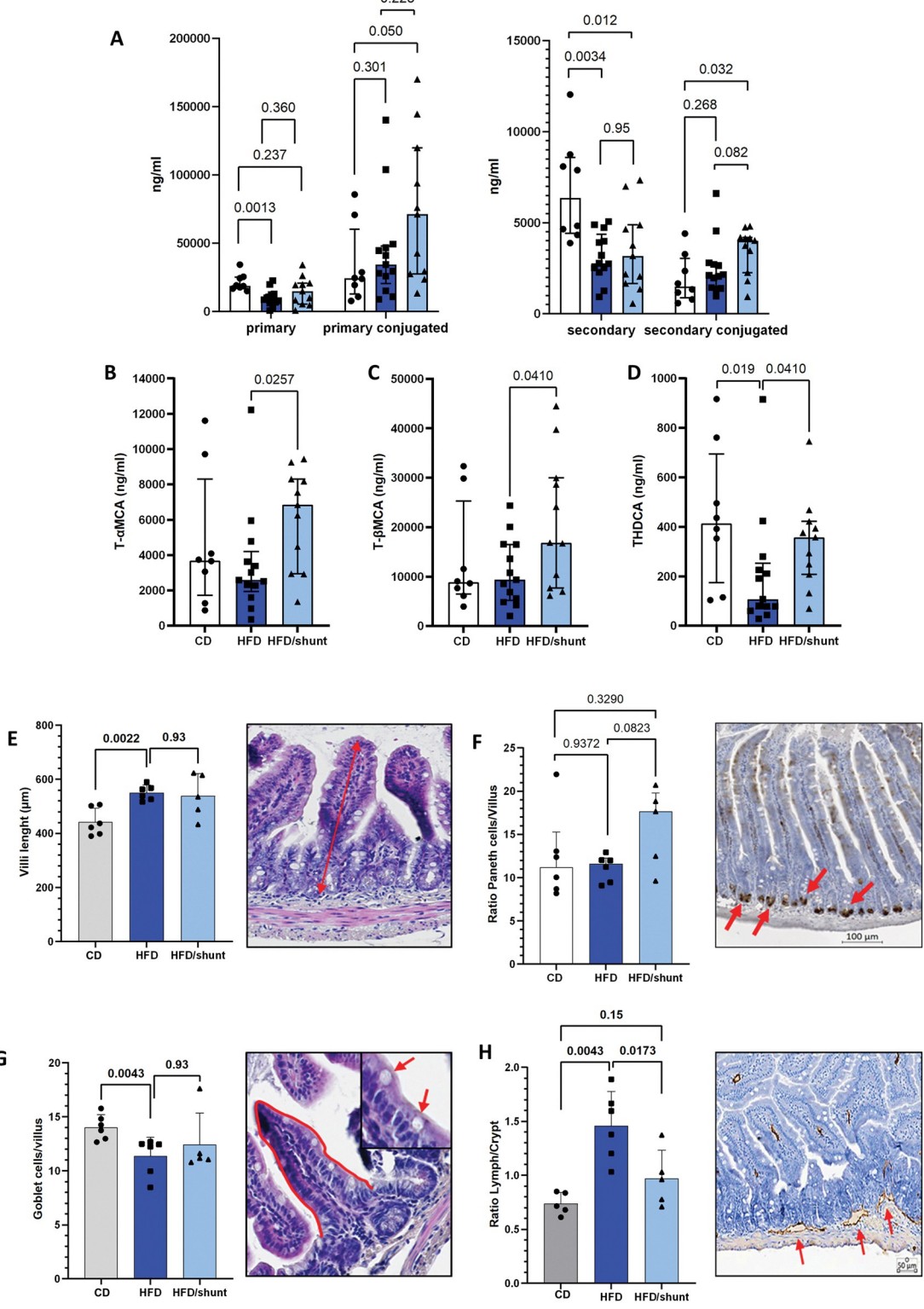

**Fig 5. Effect of portosystemic shunting on secondary bile acid metabolism and proximal bowel alterations.** Histogram showing primary and secondary bile acids (BAs) analysis in control diet-fed (CD, white, n = 8), HFD-fed mice (HFD, dark blue, n = 14) and HFD-fed mice with shunt (HFD/shunt, light blue, n = 12)(A). Portosystemic shunt regulates BAs metabolism by up-regulating the expression of T-a-MCA and T-b-MCA (B and C, respectively) and by restoring THDCA levels (D). Proximal bowel assessment of villi length (E), Paneth cells (F), Goblet cells (G) and lymphatic vessels (H) in control diet-fed (CD, white,

n = 6), HFD-fed mice (HFD, dark blue, n = 6) and HFD-fed mice with shunt (HFD/shunt, light blue, n = 6). Data are presented as median ± IQR. Comparisons between groups were performed using the Mann-Whitney U.

it opens the door to potential new therapeutic strategies with portosystemic shunting or acting solely on portal pressure.

We used a mouse model of portosystemic shunting which is easy to establish and maintain. Spleen transposition has been first described in the early 70's in rats [29], and has thus far been used mainly to preserve blood flow from the bowel in studies on liver ischemia and liver transplantation [30]. Based on the injection of labelled beads into the ileocolic vein, we estimate that such shunt can deviate about two third of the portal flow through the spleen to the systemic circulation, and reduce portal pressure to close-to-normal levels. Portosystemic shunting has therefore a dual effect: it reduces the portal pressure and direct part of the blood flow away from the liver. To uncouple the two, future mechanistic studies could use medication acting on portal pressure, including beta-blockers. Our model, however, represents a good

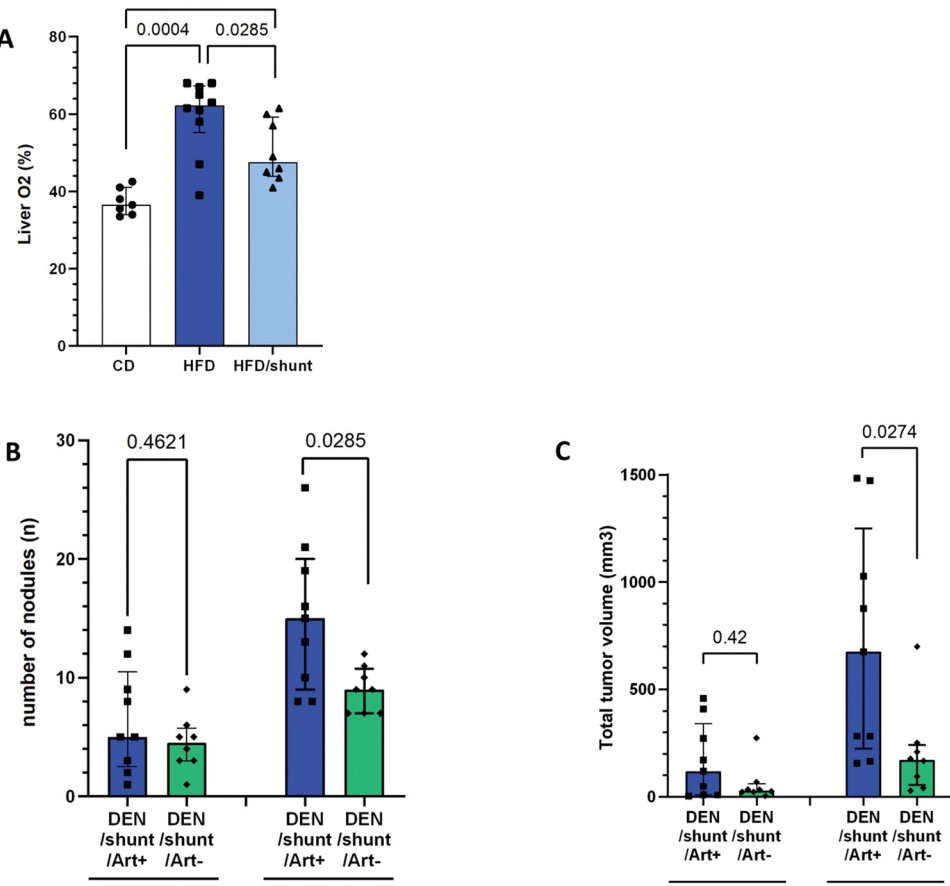

**Fig 6. Effect of portosystemic shunting on liver oxygenation.** Hyperspectral analysis of liver oxygenation in control diet-fed (CD, n = 7), HFD-fed mice (HFD, n = 10) and HFD-fed mice with shunt (HFD/shunt, n = 8) (A). In shunted mice, removal of arterial flow (Art-, n = 9) leads to a decreased carcinogenesis with less and smaller HCCs (B and C) compared to controls with preserved artery (Art+, n = 8). Data are expressed as median± IQR. Comparisons between groups were performed using the Mann-Whitney U.

approximation of TIPS used in patients, and could be utilized more widely to study vascular liver diseases.

The central message of the study is that shunting can prevent HCC carcinogenesis and divide by almost two the observed number of HCC tumours. This data is strong and reproducible in three long-term independent experiments utilizing chemically- and genetically-induced HCC mouse models.

The risk of HCC initiation and growth is linked to the level of steatosis [4, 31, 32]. We therefore observed that shunting prevents steatosis even in mice continuously fed an HFD. Steatosis is a consequence of an imbalance between lipid inflow and outflow [33]. This balance is maintained by three major factors: the uptake of circulating fatty acids, *de novo* lipogenesis, and fatty acid oxidation [34]. Accumulating evidence suggests that pathophysiological mechanisms of NAFLD are driven mainly by an increased influx of lipids into the liver and an enhanced *de novo* lipogenesis [35]. Consistently, we observed that shunting lowered the expression levels of FASn, adiponectin and ACC1 to levels close to control. Overall, this suggests a restored *de novo* lipogenesis profile, which could contributed to the observed decrease in steatosis and HCC carcinogenesis after shunting.

Also, we observed normalization of liver oxygenation after shunting, which is in line with previous data [36]. Decreased oxygen levels may be a consequence of intrahepatic hemodynamic changes, where the artery is unable to compensate for the decreased portal vein flow in all zones of the liver parenchyma. The causal link between the decreased oxygen level and the lower number of HCCs is further supported by the decreased number of tumours in shunted mice with ligated hepatic artery.

In addition, NAFLD is associated with lower concentrations of unconjugated primary and secondary bile acids in the portal vein [37]. In our study, shunting was associated with restored portal levels of T-α-MCA, T-β-MCA, and THDCA in HFD-fed mice, which could contribute to the protection against steatosis. Earlier evidence indicates that T-β-MCA can act as an antagonist for farnesoid X receptor and decrease lipid absorption [38]. Clinical and mouse data demonstrate that restoring bile acid metabolism alleviates steatosis [39]. Also, T-β-MCA has been implicated in the liver cancer immunity through its role in CXCL16 production, which promotes natural killer T cell migration within the liver, and protects from HCC [40]. In parallel, shunting was associated with a normalization of lymphangiogenesis in the ileum. Lymphatic vessels were implicated in the lipids absorption in metabolic syndrome and after small bowel surgery [26, 41].

The proposed data are clinically relevant as portal vein pressure can be modulated with TIPS and beta-blockers. Future studies could explore the impact of these strategies on NAFLD and HCC. Currently available data is conflicting. Some studies suggest a protective, and others a deleterious effect of TIPS [42], with some TIPS patients exhibiting HCCs with more aggressive features [43]. The effect of non-selective beta-blockers appears less ambiguous. Registry studies show a protective effect of beta-blocker treatment on HCC development [44].

Our data demonstrate a positive effect of portosystemic shunting on HCC carcinogenesis and steatosis when applied as a preventive measure. However, the application of shunting as a treatment later in the NAFLD disease course needs to be explored to increase the clinical relevance of these findings.

The experimental settings we observed is different of congenital portosystemic shunt, as known as Abernethy malformation, which promote the development of HCC [45–47]. These congenital shunts divert an important part of the portal flow without pre existant liver disease and portal hypertension. This may at least in part explain why we demonstrated a decrease in carcinogenesis in the HFD-fed mice model and the congenital shunts are on the contrary associated with the development of HCC.

Taken together, we demonstrate the central impact of portal haemodynamic changes on NAFLD-associated HCC carcinogenesis. According to our model, portosystemic shunting decreases portal hypertension, and prevents HCC via a decrease in steatosis and liver oxygenation levels.

## Supporting information

**S1 Fig. Shunting does not alter vascular reactivity of the aorta.** Dose-response curve established by applying increasing concentrations of KCl (10–100 mmol/L) (A) or U46619 (10–9 to 10–5 mol/L)(B) in aortic vessel isolated from control-diet mice (CD; n = 4; white curve), high-fat diet mice (HFD; n = 9; dark blue curve) and HFD/shunt mice (n = 11; light blue curve). Relaxation dose-response curves established by applying increasing concentrations of acetylcholine (Ach; 10–8 mol/L to 10–3 mol/L) (C) and Sodium Nitroprusside (SNP; 10-9mol/L to 10–3 mol/L) after U46619 pre-contraction (1 μM) (D), on aortic segments isolated from control-diet mice (CD; n = 4; white curve), high-fat diet mice (HFD; n = 9; dark blue curve) and HFD/shunt mice (n = 11; light blue curve).
(TIF)

**S2 Fig. Shunting does not affect heart rate and blood pressure profiles.** Heart rate (A) and arterial pressure (B) of CD mice (n = 8), HFD mice (n = 8) and HFD/shunt mice (n = 8).
(TIF)

**S3 Fig. Shunting reduced the number of HCC.** MicroCT sections of livers at 28 weeks and 36 weeks and pictures of livers at 40 weeks (HFD and HFD/Shunt).
(TIF)

**S1 Table. Primers sequences.**
(TIF)

**S1 File. MicroCT movies of HFD and HFD/Shunt mice at 36weeks (HFD.wmv and HFD-shunt.wmv).**
(ZIP)

## Acknowledgments

We thank the teams of the Genomics, Histology, and Bioimaging core facilities, as well as the small animal preclinical imaging platform and the phenotyping of small animal facility of the Faculty of Medicine (University of Geneva). The authors sincerely acknowledge Beata Kusmider for the meaningful proofreading of the manuscript.

## Author Contributions

**Conceptualization:** Andrea Peloso, Stéphanie Lacotte, Brenda Kwak, Laura Rubbia-Brandt, Christian Toso.

**Formal analysis:** Andrea Peloso, Stéphanie Lacotte, Beat Moeckli, Matthieu Tihy, Aurélie Hautefort.

**Funding acquisition:** Andrea Peloso, Stéphanie Lacotte, Christian Toso.

**Investigation:** Andrea Peloso, Stéphanie Lacotte, Quentin Gex, Florence Slits, Graziano Oldani, Matthieu Tihy, Aurélie Hautefort.

**Methodology:** Andrea Peloso, Stéphanie Lacotte, Aurélie Hautefort.

**Project administration:** Christian Toso.

**Supervision:** Stéphanie Lacotte, Brenda Kwak, Laura Rubbia-Brandt, Christian Toso.

**Writing – original draft:** Andrea Peloso, Stéphanie Lacotte.

**Writing – review & editing:** Andrea Peloso, Stéphanie Lacotte, Beat Moeckli, Matthieu Tihy, Aurélie Hautefort, Brenda Kwak, Christian Toso.

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
