## [Decision Letter · Decision Letter 0]

29 Nov 2023

PONE-D-23-16913Portosystemic shunting prevents hepatocellular carcinoma in non-alcoholic fatty liver disease mouse modelsPLOS ONE

Dear Dr. Lacotte,

Thank you for submitting your manuscript to PLOS ONE. After careful consideration, we feel that it has merit but does not fully meet PLOS ONE’s publication criteria as it currently stands. Therefore, we invite you to submit a revised version of the manuscript that addresses the points raised by the reviewer.

We look forward to receiving your revised manuscript.

Kind regards,

Matias A Avila, Ph.D.

Academic Editor

PLOS ONE

Journal Requirements:

3.We note that the grant information you provided in the ‘Funding Information’ and ‘Financial Disclosure’ sections do not match.

"NO"

Reviewers' comments:

Reviewer's Responses to Questions

**Comments to the Author**

1. Is the manuscript technically sound, and do the data support the conclusions?

Reviewer #1: Partly

2. Has the statistical analysis been performed appropriately and rigorously? 

Reviewer #1: Yes

3. Have the authors made all data underlying the findings in their manuscript fully available?

Reviewer #1: Yes

4. Is the manuscript presented in an intelligible fashion and written in standard English?

Reviewer #1: Yes

5. Review Comments to the Author

Reviewer #1: The authors presented the effect of portosystemic shunting on the prevention of HCC development in NAFLD mouse models. The phenomena, reduction of portal pressure and decreased incidence of HCC development are clearly demonstrated; however, the direct association of the modification of gut-liver axis by the shunting on HCC prevention is still unclear.

Therefore, authors need to demonstrate the modification of gut-liver axis by showing for e.g. microbiota changes and the association of the shunting on bile acid and/or oxygenation.

Thus, this manuscript needs to be revised according to the concerns described above before further consideration on its publication.

6. PLOS authors have the option to publish the peer review history of their article (what does this mean?). If published, this will include your full peer review and any attached files.

Reviewer #1: No

---

## [Author Response · Author response to Decision Letter 0]

8 Dec 2023

The answer to reviewer comments can be found in the PointByPointResponse.docx file.

---

## [Decision Letter · Decision Letter 1]

10 Dec 2023

Portosystemic shunting prevents hepatocellular carcinoma

in non-alcoholic fatty liver disease mouse models

PONE-D-23-16913R1

Dear Dr. Lacotte,

We’re pleased to inform you that your manuscript has been judged scientifically suitable for publication and will be formally accepted for publication once it meets all outstanding technical requirements.

Kind regards,

Matias A Avila, Ph.D.

Academic Editor

PLOS ONE

Additional Editor Comments (optional):

Reviewers' comments:

Reviewer's Responses to Questions

**Comments to the Author**

1. If the authors have adequately addressed your comments raised in a previous round of review and you feel that this manuscript is now acceptable for publication, you may indicate that here to bypass the “Comments to the Author” section, enter your conflict of interest statement in the “Confidential to Editor” section, and submit your "Accept" recommendation.

Reviewer #1: (No Response)

2. Is the manuscript technically sound, and do the data support the conclusions?

Reviewer #1: (No Response)

3. Has the statistical analysis been performed appropriately and rigorously? 

Reviewer #1: (No Response)

4. Have the authors made all data underlying the findings in their manuscript fully available?

Reviewer #1: (No Response)

5. Is the manuscript presented in an intelligible fashion and written in standard English?

Reviewer #1: (No Response)

6. Review Comments to the Author

Reviewer #1: (No Response)

7. PLOS authors have the option to publish the peer review history of their article (what does this mean?). If published, this will include your full peer review and any attached files.

Reviewer #1: No

---

## [Editor Report · Acceptance letter]

18 Dec 2023

PONE-D-23-16913R1 

PLOS ONE

Dear Dr. Lacotte, 

I'm pleased to inform you that your manuscript has been deemed suitable for publication in PLOS ONE. Congratulations! Your manuscript is now being handed over to our production team.

Kind regards, 

on behalf of

Dr Matias A Avila 

Academic Editor

PLOS ONE